# An Approximation of Heart Failure Using Cardiovascular Simulation Toolbox

**DOI:** 10.3390/biomimetics4030047

**Published:** 2019-07-10

**Authors:** Gabriela Ortiz-León, Heiner José Barrantes-Vargas, Manuel Arguedas-Sandí, Jose Ángel Pacheco-Chaverri, Marta Vílchez-Monge

**Affiliations:** SIBILA Lab. Department of Mechatronics Engineering, Tecnológico de Costa Rica, Cartago 30101, Costa Rica

**Keywords:** cardiovascular simulation toolbox, heart failures, heart simulation

## Abstract

In this paper, we present the simulation of 5 different heart failures with the help of the Cardiovascular Simulation Toolbox (CVST) proposed by O. Barnea et al. at Tel-Aviv University. This is a modified version of the CVST, proposed by G.Ortiz; here, we show that the pathological failures can be covered by this tool. We varied the value of the tool blocks, included the results of the hemodynamic parameters and the P-V loop curves for each disease and compared them to the medical data to prove the effectiveness of the simulation. Based on these changes, we achieved an effective simulation of the following heart failures in the CVST: Diastolic Heart Failure (DHF), Systolic Heart Failure (SHF), Right Ventricle Heart Failure (RVHF), Low Output Heart Failure (LOHF) and High Output Heart Failure (HOHF).

## 1. Introduction

Circuits not only allow us to develop complex electronic systems, they also help us in the simulation and modeling of physical systems. The cardiovascular system is one the many examples which can be represented through an electric equivalent system. To set some examples: the heart operates like a hydraulic pump whose blood pressure output behaves like a variable tension source; the heart beats produce sudden shifts in the internal pressure, which makes the tissue in our heart expand as a circuit would react during its transitory phase; veins can be represented electrically as a serial line with complex impedance. Due to these circumstances, the circulatory human system can be represented and simulated by an equivalent electric system.

In this paper, the human heart is simulated alongside some other heart failures. These simulations were implemented in MATLAB/Simulink^®^R2015a using the Cardiovascular Simulation Toolbox (CVST) tool which was originally created by O. Barnea [1] and was later updated in Reference [2]. Only for a matter of clarity, in this paper, we are calling this updated version of the toolbox CVST-2. This tool has a series of programmed blocks that represent different sections of the circulatory system. To achieve a successful simulation of different heart failures, the obtained values were contrasted with the results of other existing models and medical data gathered in specialized laboratories.

Simulations of the human body can be useful for developing medical equipment, since complex devices such as VAD (Ventricular Assistance Device) or pacemakers could be better developed by taking in account this extra layer of information in their design stages. Tools, such as the one presented in this paper, could benefit the design process by predicting and estimating the human body behavior after a device has been implanted.

Heart failures are caused by a dysfunction of the heart, which denies an adequate blood flow and correct transport of oxygen and nutrients necessary to the metabolism of different tissues [3]. They can be divided in two types [4]:Depending of the output in the ventricles: “High-Output” and “Low-Output”.Depending of their location: “Left-Sided Systolic,” “Left-Sided Diastolic” and “Right-Sided Right-Ventricle.”

Heart failures could also be explained in terms of their causes and effects. Since it was necessary to observe the effect of different diseases in the cardiovascular system, the model of a healthy condition [2] was modified to obtain different failures; this was accomplished due to the ease and flexibility of the programmed blocks of the CVST-2.

## 2. Simulation Model

The CVST-2 tool is a series of programmed blocks which together describes the behavior of the different parts of the cardiovascular system, inside them are a series of electrical connections of different active and passive components, signals and mathematical operations. This tool is divided in five categories:Heart: Contains auricles, ventricles and valves.Measurements: Designed to obtain and visualize different parameters in the system.Veins: Contains power transmission-based circuits.Oxygen Transport: Contains blocks which calculate the oxygen concentration and the gas exchange between blood and the tissues.Other: Groups the other components that don’t fit the description of the previous categories, such as resistances and flow valves [2].

The complete model sketch of the cardiovascular system in a “healthy condition” can be appreciated in Figure 1.

The “healthy condition” refers to the simulation obtained of a healthy person, by using the CVST-2 tool. In addition, this simulation was previously validated by comparing its results with standard medical values of a healthy person [2]. This comparison between the simulation and the medical data is shown in Table 1. Since the simulation values are inside the established limits, this confirms the validity of the simulation for a healthy heart.

Moreover, this tool prints the curves or functions of the different hemodynamic parameters shown in Table 1. An example of this, which is not explicit in the previous table, is shown in the Figure 2, where a P-V loop can be observed. This specific graph allows us to identify a diversity of conditions, such as changes in the elastance, as well as hemodynamic parameters like the Left Ventricular End-Diastolic Volume (LVEDV), Left Ventricular End-Systolic Volume (LVESV) and Left Ventricular Stroke Volume (LVSV), among others [5]. With these parameters is possible to determine whether the heart is healthy or presents a failure condition. It should be noted that the data used for the reference values in Table 1, were obtained from [6,7,8,9,10].

## 3. About the Heart Failures

The simulated heart failures presented in this article are the following:High-Output Heart Failure (HOHF): It happens when the body needs high levels of blood, due to that the heart beat faster than usual [11]. Causes of HOHF could be: Anemia, hipoertiroidism, arteriovenous fistula, Beriberi illnesses and Paget’s in Reference [12]. Lobato [13] mentions that HOHF is the result of an increase in the diastolic pressure and the volume on the left ventricle. Some effects are: marked vasodilatation, pulse increase, water and salt retention, renal fluid diminution and activation on the neuroendocrine system [13]. Also, heart function could be supra normal but this will be inadequate for the high metabolic necessities [4].Low-Output Heart Failure (LOHF): It happens due to cardiac disorders that damages the adequate bombing function, due to diastolic and systolic dysfunctions. LOHF can produce damage to the valves, arrhythmia, vasoconstrictions and cyanotic and cold extremities. The cardiac output is normal but it tends to decrease when the body undergoes physical exertion [4].Systolic Heart Failure (SHF): It is the heart’s incapacity of bombing blood at its normal speed. The ejection fraction in the left ventricle plays an important role when diagnosing it, due to its significant diminution. This is a lethal sickness hence, deaths are very spontaneous [14]. Some physical effects are: the cavity of the left ventricle dilates, although the thickness of the wall maintains in a similar way; the contractile function of the heart seems to reduce. Due to that fact, the ejection fraction reduces and so the systolic volume.Diastolic Heart Failure (DHF): It relates with the difficulty of filling one or both ventricles, as also, the camera of the ventricle does not accept the appropriate level of normal blood pressure. In this case, the quantity of blood that is bombed every beat in the left ventricle is normal [14]. Some risk factors are: diabetes, hypertension, age, obesity and coronary artery sickness. Physically, the cavity of the left ventricle maintains the same or reduces but on the other hand, the thickness of the wall increases significantly.Right-Ventricle Heart Failure (RVHF): It is a consequence from the damage resulted in the left side of the heart, the increase of blood pressure transfers to the lungs, thus going back to the right side of the heart. The blood accumulates in the veins, each time that the right ventricle loses pumping power. Some effects of RVHF are: inflammations in ankles and legs [15].

## 4. Results

The results of different simulations with input values in Table 2, are shown in Table 3. This is a comparative table between the output of the different parameters in the conditions previously described and the healthy condition.

Moreover, the graphs of P-V loops will be shown for a further and more extensive comprehension of the simulated conditions. This type of graph helps us describe the behavior of the heart such as its effectiveness to pump blood and the internal efforts, among other characteristics. Figure 3 shows the P-V loop of the HOHf simulation in the CVST-2; similarly, Figure 4 shows in the same axes the P-V loop of the LOHF simulation.

It should be noted, that depending of each simulation the curve in the graph shows displacements in the horizontal axis, which represents the amount of blood in the chamber. Also, the increase or decrease in height represents the rise or fall of pressure; which is also useful to describe health conditions in patients. For the first two graphs, although the curve stays centered between the volumes of 80 mL and 180 mL; we can see that each one has a different height, which means that the simulations predict conditions of high and low pressure.

Figure 5 shows the P-V loop of the SHF simulation; Figure 6 shows the same type of curve and in the same axes used before for the DHF simulation. Finally, Figure 7 shows the P-V loop for the RVHF simulation. The conditions showed in the three previous graphs, shows that the curves are moved along the horizontal axis; this indicate that the simulations are related to conditions in which the structure of the heart (wall’s thickness and chamber’s cavities) could be altered.

## 5. Discussion

Given the results shown in the previous section, we analyzed the data and compared each simulation with the general characteristics, as well as some parameters from different authors, to evaluate if the simulated condition is representative of each heart failure.

HOHF: As shown in Figure 3, there’s a displacement to the right of the P-V loop curve, which indicates a raise of the volume in the left ventricle of the heart. In addition, the increase in pressure of curve represents a condition of exertion in the heart muscles, therefore, there’s an expected higher output of blood. This is demonstrated by the data shown in Table 2, where the CO and HB increase relative to the healthy condition; both conditions are typical in this type of failure. Furthermore, the expected values for the cardiac output and heart beats are around 8 L/min [16] and between 85 and 105 bpm [17]; in both cases, the values are approximately the same which demonstrates a consequent behavior with failure description.

LOHF: In Figure 4, there’s a light displacement of the curve of approximately 10 mL to the right, as well as a beat pressure diminution compared to the one shown in Figure 2. This condition indicates that, even though there’s a raise in volume, the heart suffers from a deficiency in the force excelled to pump blood correctly. Can also be observed in the output values of Table 2; this phenomenon in addition to the HB decrease, proves that the simulation has the same general characteristics as the description of the LOHF.

SHF: In the P-V loop curve shown in Figure 5 a significantly increase in volume in the left ventricle can be observed, as well as a decrease in pressure. These characteristics are caused by exhaustion in the heart chamber’s walls, which cause a bigger cavity and lower pressure due to the thinness of the walls [14]. Additionally, some of the characteristic values for this failure are: LVEDV of 192 ± 10 mL, LVESV of 137 ± 9 mL and LVEF of 31 ± 2% [14], in the Table 2, these values are respectively: LVEDV is 180.98 mL, LVESV is 136.54 mL and LVEF is 24.55%. With these values and the characteristics of the P-V loop, there’s enough evidence to that the general behavior of the simulation is similar to the description of SHF.

DHF: In Figure 6, the P-V loop shows a displacement to the left, which traduces to less volume in the ventricle relative to the healthy condition; this movement alongside the decrease pressure in the curve produces a reduction of the CO, as it is shown in the Table 2. As mentioned in Reference [14], this condition is caused by the thickening of the heart walls, which reduces its pumping capacity and its maximum blood volume. In addition, some theoretical range values for the hemodynamic parameters are presented in Reference [14]: LVEDV of 87 ± 10 mL, LVESV of 37 ± 9 mL and LVEF of 60 ± 2%. In the Table 2, these variables have a value of: LVEDV 95.97 mL, LVESV 45.71 mL and LVEF 52.36%; which then again, proves that the simulation for this heart failure is successfully represented.

RVHF: In Figure 7, the decrease in the pumping pressure for the systolic phase, reflects the reduction of the CO shown in Table 2. In addition, this table shows the average reduction of the right ventricle pressure, which is an important characteristic of this type of failure. It should be noted that with the information presented, the damage in the right ventricle also affects the other parts of the heart and the behavior is similar to the description given in Reference [15].

## 6. Input Parameters

Most heart failures produce damages such as the thickening of heart walls, diminution of the heart chamber’s contraction, reduction of the elastic tissues walls and increments of the heart valve’s resistance. These effects are meaningful in the simulation of heart failures, since those are a starting point in determining the input parameters values.

The modifications for the simulation model variables are shown in Table 3. Among the variables changes, it can be noted that the heart beat adjusts between the failures, as the description suggests. The chamber’s elastance was changed to represent damage or loss of elasticity in the tissues. Moving along these lines, the resistance of the valves and the initial volume were changed to affect the ejection fraction. It should be stressed, that these changes followed the description given before of each condition.

## 7. Conclusions

The CVST-2 tool, not only provides an adequate simulation of the hemodynamic parameters of a healthy person but it also allows the simulation of the heart conditions due to different failures. Due to the ease of these parameter’s manipulation, we achieved, through simulation, the general behavior of five different heart failures. With this, we can provide a more precise description of the heart’s behavior to study in vitro the effect of adding different devices or components in the cardiovascular system, such as artificial cardiac pacemaker or a ventricular assist device. It should be also noted that this tool provides help for students in the biological area, as they can see and manipulate a heart’s condition in a simulation, instead of having to look for records of previous patients. Which gives this tool a great importance in development of medical devices as well for academic purposes.

## Figures and Tables

**Figure 1 biomimetics-04-00047-f001:**
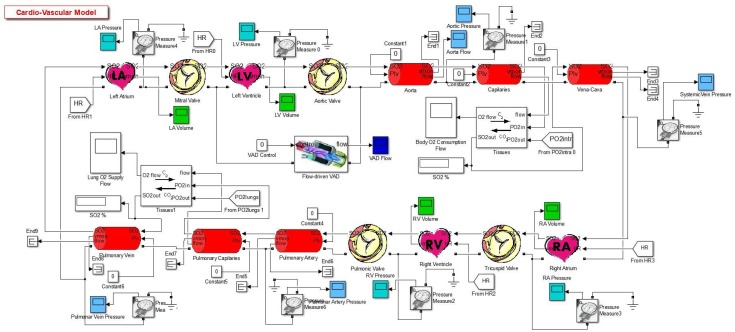
Complete model [2].

**Figure 2 biomimetics-04-00047-f002:**
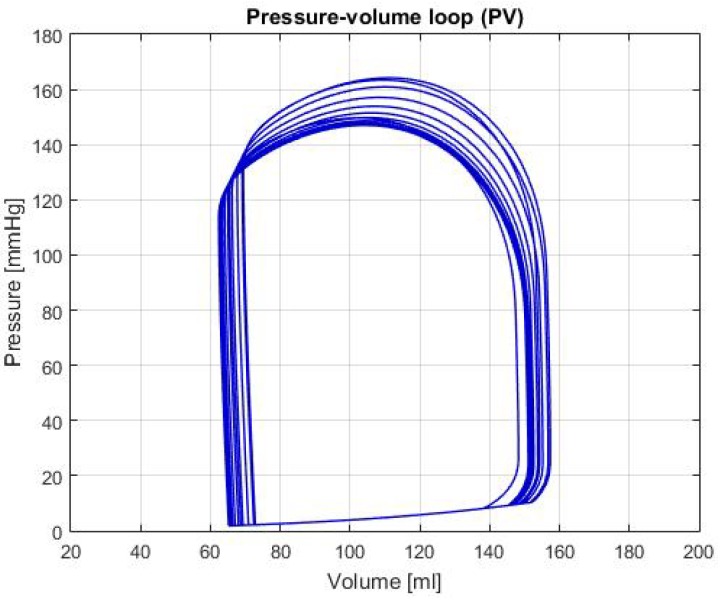
Graph of P-V loop curves in the left ventricle with a healthy condition.

**Figure 3 biomimetics-04-00047-f003:**
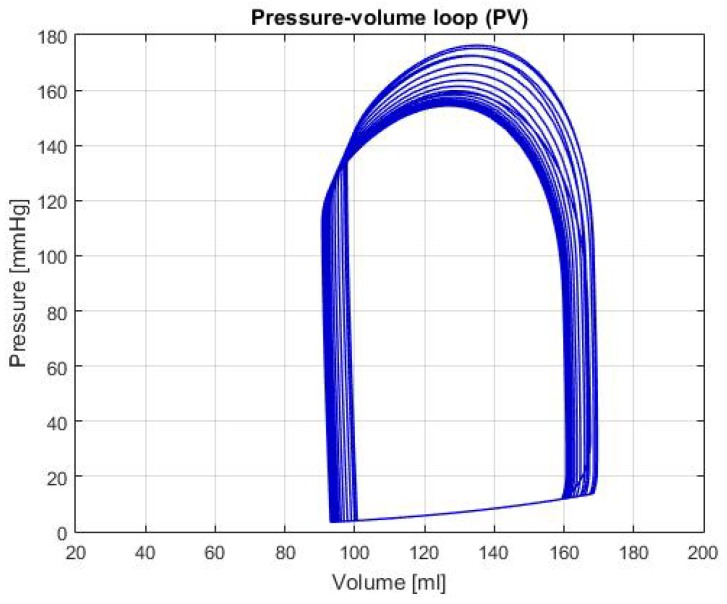
Graph of P-V loop curves with High Output Heart Failure (HOHF).

**Figure 4 biomimetics-04-00047-f004:**
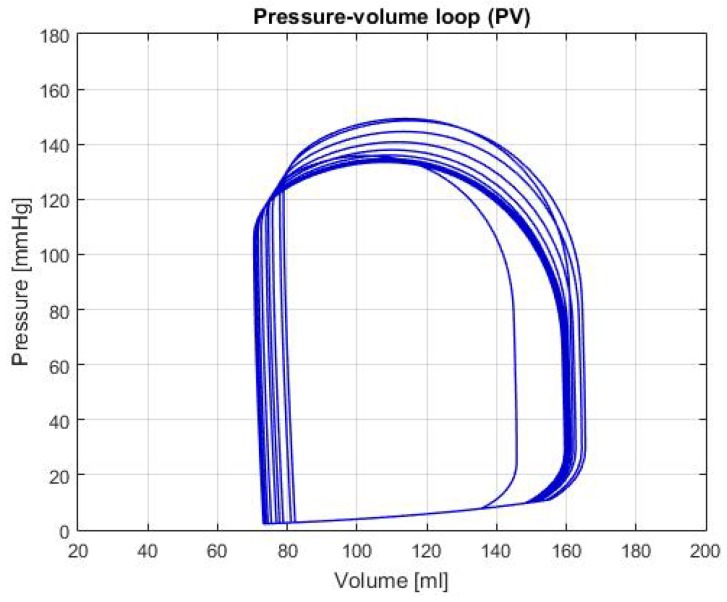
Graph of P-V loop curves with Low Output Heart Failure (LOHF).

**Figure 5 biomimetics-04-00047-f005:**
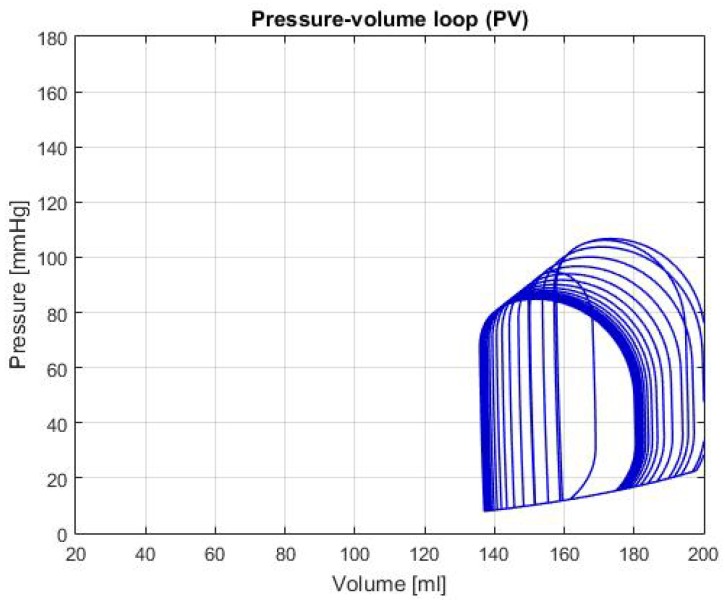
Graph of P-V loop curves with Systolic Heart Failure (SHF).

**Figure 6 biomimetics-04-00047-f006:**
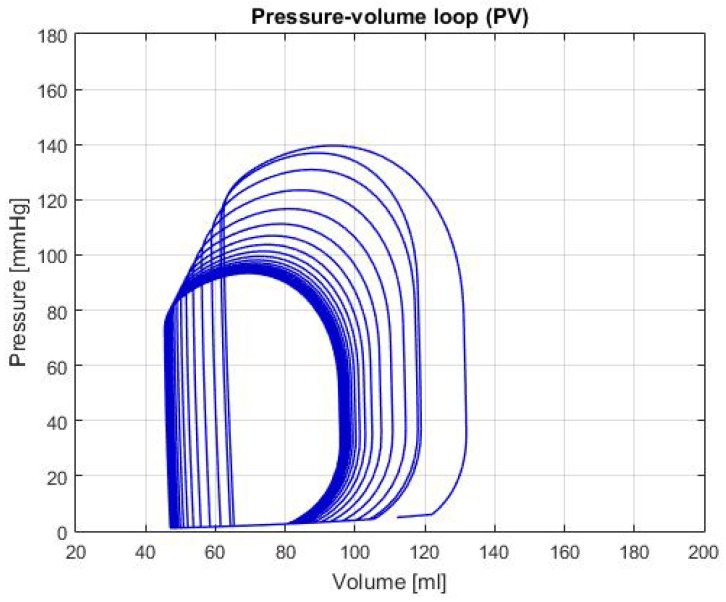
Graph of P-V loop curves with Diastolic Heart Failure (DHF).

**Figure 7 biomimetics-04-00047-f007:**
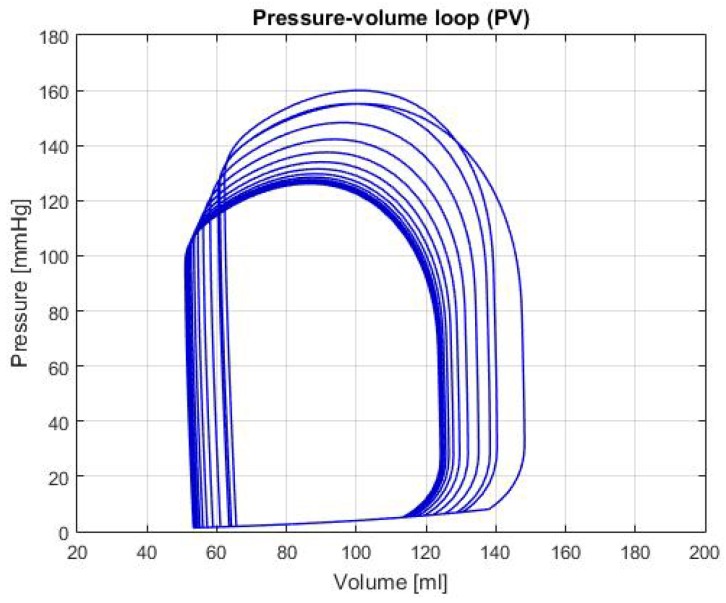
Graph of P-V loop curves with Right Ventricle Heart Failure (RVHF).

**Table 1 biomimetics-04-00047-t001:** Comparison of hemodynamic parameters between a healthy person and the Cardiovascular Simulation Tool (CVST-2) simulation.

Parameter	Units	Reference Value	Normal	
			*Min*	*Max*
Heart Rate	bpm	48–105 [6]	75	-
Cardiac output	L/min	4.0–8.0 [6]	6.7154	-
Left Ventricular Volume	mL	16–239 [7]	62.5976	152.1365
Right Ventricular Volume	mL	50–160 [6]	78.7093	152.6721
Left Atrial Volume	mL	25.3–109.7 [9]	19.3332	102.0856
Right Atrial Volume	mL	26.4–121.5 [9]	7.3287	79.7041
Arterial Pressure	mmHg	60–140 [10]	68.47	144.3845
Left Ventricular Pressure	mmHg	0–140 [10]	1.8499	147.8072
Left Atrial Pressure	mmHg	4.0–12 [6]	0.4368	30.1822
Right Ventricular Pressure	mmHg	2.0–30 [6]	2.1612	41.7744
Right Atrial Pressure	mmHg	2.0–6.0 [6]	0.2426	21.4079
LV Stroke Volume	mL	60–100 [6]	89.5389	-
RV Stroke Volume	mL	60–100 [6]	73.9628	-
LV End-Diastolic Volume	mL	120 (65–239) [8]	152.1365	-
LV End-Systolic Volume	mL	50 (16–143) [8]	62.5976	-
RV End-Diastolic Volume	mL	100–160 [6]	152.6721	-
RV End-Systolic Volume	mL	50–100 [6]	78.7093	-
LV Ejection Fraction	%	59.9 (18–76) [8]	58.8543	-
RV Ejection Fraction	%	40–60 [6]	48.4455	-

**Table 2 biomimetics-04-00047-t002:** Values of the input variables for the simulation of different conditions.

Variable	Normal	SHF	RVHF	LOHF	HOHF	DHF
Heart Rate	75	75	75	65	100	75
LV Emax	2.5	1	2.8	2	2.9	2.5
LV P-V graph intercepted volume	15	60	15	15	50	15
LV End-Diastolic Volume	112	137	120	112	132	112
MV Resistance	0.02	0.05	0.03	0.02	0.03	0.05
AV Resistance	0.002	0.005	0.003	0.006	0.008	0.005
RV Emax	0.85	0.35	0.3	0.85	0.85	0.85
RV P-V graph intercepted volume	50	60	50	50	50	50
RV Initial Volume	80	65	80	80	80	80
TV Resistance	0.02	0.05	0.03	0.03	0.03	0.05
PV Resistance	0.002	0.005	0.008	0.004	0.003	0.005
LA Emax	1	0.8	1	1	1	0.8
LA BFR	0.1	0.15	0.1	0.1	0.1	0.15
RA Emax	1	0.8	1	1	1	0.8
RA BFR	0.1	0.2	0.1	0.1	0.1	0.2

**Table 3 biomimetics-04-00047-t003:** Output hemodynamic values for different heart failures in the CVST-2.

Parameter		Units	Normal		SHF		RVHF		LOHF		HOHF		DHF	
			*Min*	*Max*	*Min*	*Max*	*Min*	*Max*	*Min*	*Max*	*Min*	*Max*	*Min*	*Max*
HR	Heart Rate	bpm	75	-	75	-	75	-	65	-	100	-	75	-
CO	Cardiac output	L/min	6.7154	-	3.3494	-	5.5054	-	5.879	-	7.0117	-	3.7728	-
LVV	Left Ventricular Volume	mL	62.5976	152.1365	135.793	180.4515	50.9441	124.3494	70.7789	161.2252	90.635	160.7519	45.439	95.7433
RVV	Right Ventricular Volume	mL	78.7093	152.6721	117.8721	156.0561	128.5198	189.2988	78.5933	154.2769	77.3493	134.5208	74.253	115.8001
LAV	Left Atrial Volume	mL	19.3332	102.0856	27.195	83.431	18.7101	93.7377	20.826	114.4978	16.2727	80.9615	29.0205	84.0079
RAV	Right Atrial Volume	mL	7.3287	79.7041	11.3458	50.6052	11.3257	82.8504	9.0201	89.2757	6.2522	62.3506	9.909	50.3329
AP	Arterial Pressure	mmHg	68.47	144.3845	44.4787	81.3584	59.2085	122.4431	60.1552	129.1583	75.4493	146.3819	46.3871	89.3429
LVP	Left Ventricular Pressure	mmHg	1.8499	147.8072	7.9626	84.8965	1.3463	126.253	2.243	134.5208	3.4325	154.4136	1.1235	93.3702
LAP	Left Atrial Pressure	mmHg	0.4368	30.1822	0.5485	35.1587	0.4093	31.3536	0.4849	37.0309	0.3594	31.2227	0.5907	36.8683
RVP	Right Ventricular Pressure	mmHg	2.1612	41.7744	4.5958	30.7118	5.5388	37.0351	2.1566	41.3696	2.0972	41.3268	1.961	33.6142
RAP	Right Atrial Pressure	mmHg	0.2426	21.4079	0.2441	21.5873	0.2674	28.7435	0.2673	26.1614	0.1876	23.3241	0.2349	19.7209
LVSV	Left Ventricular Stroke Volume	mL	89.5389	-	44.6585	-	73.4053	-	90.4463	-	70.117	-	50.3043	-
RVSV	Right Ventricular Stroke Volume	mL	73.9628	-	38.184	-	60.779	-	75.6836	-	57.1715	-	41.5471	-
LVEDV	Left Ventricular End-Diastolic Volume	mL	152.1365	-	180.4515	-	124.3494	-	161.2252	-	160.7519	-	95.7433	-
LVESV	Left Ventricular End-Systolic Volume	mL	62.5976	-	135.793	-	50.9441	-	70.7789	-	90.635	-	45.439	-
RVEDV	Right Ventricular End-Diastolic Volume	mL	152.6721	-	156.0561	-	189.2988	-	154.2769	-	134.5208	-	115.8001	-
RVESV	Right Ventricular End-Systolic Volume	mL	78.7093	-	117.8721	-	128.5198	-	78.5933	-	77.3493	-	74.253	-
LVEF	Left Ventricular Ejection Fraction	%	58.8543	-	24.7482	-	59.0315	-	56.0993	-	43.6181	-	52.5409	-
RVEF	Right Ventricular Ejection Fraction	%	48.4455	-	24.4681	-	32.1075	-	49.057	-	42.5001	-	35.8783	-

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
