# Peer review of "An Approximation of Heart Failure Using Cardiovascular Simulation Toolbox"

_biomimetics, 2019, doi:10.3390/biomimetics4030047_

Round 1
Reviewer 1 Report
This is a nice implementation of the CVS toolbox and its modified version. The paper shows that indeed the pathological states are covered by the inherent models in the toolbox. this is important but what the paper lacks is conclusions regarding further applications in the classroom and in the clinic. It should have a message on top of the actual mathematical value.
Author Response
We added a small explanation about posible uses of the results
We did not understand the last sentence in your review
Reviewer 2 Report
The authors simulated five different types of hemodynamics including healthy and diseased heart conditions using electrical circuit analogies, and this approach may contribute to cost reduction and shortening the duration of the design process of cardiac medical devices. The paper provides interesting data, but it needs a considerable revision to be acceptable for publication.
Major comments:
1. Although the authors successfully represented several types of hemodynamics, the authors use existing tool and the description of originality is inadequate. It is important to clarify the difference between the present study and previous studies. The authors should describe the originality and the novelty of your study.
2. I could not find the Discussion section in the manuscript and I think that the results of the manuscript are not well discussed. The authors should follow the submission guideline.
Minor comments:
1. I do not recommend using abbreviations in the Abstract, and the authors should spell out each of abbreviations the first time it appears in the main text.
2. There are quite some typographical errors in the manuscript. Please correct the spelling carefully.
I hope these comments will be helpful.
Author Response
Major comments
This is a modified version of CVST, the paper shows that indeed the pathological states are covered and can be simulated by the inherent models in the toolbox
The submition guideline was followed as recommended
Minor comments
Recommendation followed
The spelling has been revised